# Flowing Through States: Neural ODE Regularization for Reinforcement Learning

**Mohamed Ghanem**[1]     **Bernd Finkbeiner**[1,2]

[1]CISPA Helmholtz Center for Information Security
[2]Technical University of Munich
`mohamed.ghanem@cispa.de, finkbeiner@cispa.de`

## Abstract

Neural networks applied to sequential decision-making tasks typically rely on latent representations of environment states. While environment dynamics dictate how semantic states evolve, the corresponding latent transitions are usually left implicit, creating a potential misalignment between the two. We propose to model latent dynamics explicitly by drawing an analogy between Markov decision process (MDP) trajectories and ordinary differential equation (ODE) flows: in both cases, the current state fully determines its successors. Building on this view, we introduce a neural ODE-based regularization method that enforces latent embeddings to follow consistent ODE flows, thereby aligning representation learning with environment dynamics. Although broadly applicable to deep learning agents, we demonstrate its effectiveness in reinforcement learning by integrating it into Actor-Critic algorithms. Our approach yields major performance gains across various standard Atari benchmarks for A2C and gridworld environments for PPO.

## 1 Introduction

A central challenge in machine learning is bridging the gap between an object's semantic meaning and its latent representation. Because neural networks operate on learned embeddings rather than direct semantics, representation learning has largely focused on designing processes that faithfully encode local object properties. For instance, convolutional neural networks (LeCun et al., 1989) incorporate inductive biases such as translation equivariance, spatial locality, and approximate invariance to scale and rotation. These architectural choices encode object-level regularities, ensuring that embeddings reflect structural properties intrinsic to individual objects.

While such local representations are powerful for perception tasks, *sequential decision-making* introduces a different challenge: the need for a more *global* understanding of how objects and states relate to one another over time. In this setting, the relevant inductive biases emerge not from isolated objects but from the dynamics that connect them. For example, in the context of Markov Decision Processes (MDPs), the latent embeddings of a state and its successor should be consistently related by the transition dynamics. Concretely, if a transition rule $R$ connects state $s_1$ to $s_2$, then their embeddings should satisfy a relation of the form:

$$h(s_2) = g(h(s_1), R),$$

where $h(\cdot)$ denotes the embedding function, and $g$ is an arbitrary function. While the existence of such a mapping is trivial in principle, the structural properties it imposes on the latent space, such as smoothness, consistency, and determinism, are far from trivial and are crucial for reasoning tasks.

This paper proceeds from the intuition that embeddings of *semantic trajectories* can be understood as discretizations of continuous latent flows. In other words, each trajectory in the semantic space should correspond to a smooth path in the latent space. We argue that regularizing latent embeddings to respect this path structure captures an inherent property of transition dynamics, and enhances the model's ability to learn the task on a more global level. To operationalize this idea, we define latent flows using neural ordinary differential equations (neural ODEs) (Chen et al., 2018), which guarantee unique continuous trajectories under mild regularity assumptions such as Lipschitz continuity (Coddington & Levinson, 1955). In reasoning contexts, this uniqueness naturally subsumes

*the Markov property*: an initial condition (i.e., a state) completely determines the flow path of subsequent conditions.

However, directly using neural ODEs for inference is impractical: their reliance on numerical integration makes them significantly slower than standard forward passes, and their application to sequential inference is further complicated by the discontinuities introduced by evolving semantic states (Du et al., 2020; Jia & Benson, 2019; Rubanova et al., 2019). To overcome these limitations, we propose to train the agent's semantic embedder to *mimic* the flows of a neural ODE through an alignment penalty. This approach enables the learned embeddings to inherit the topological structure of smooth ODE flows, while avoiding the computational and design burdens of ODE-based inference. Our method thus combines the expressivity of continuous-time dynamics with the efficiency of conventional neural architectures. Moreover, it adds a layer of global guidance to the agent in the form of a neural ODE that learns to model the latent agent-environment dynamics in an unsupervised fashion.

The relevance of this perspective is particularly pronounced in *discrete-state* MDPs. In continuous-state environments, the inherent continuity of the state space naturally induces smoothness in the latent representations: small changes in the input state often correspond to small changes in the embedding. By contrast, in discrete domains the semantic space consists of isolated states with no *a priori* notion of proximity or smooth transitions. As a result, continuity must be imposed in the latent space rather than inherited from the state space itself. Embedding discrete trajectories as smooth latent flows therefore provides a principled way to recover structural regularities that are otherwise absent, enabling latent dynamics to reflect the transition constraints of the underlying MDP.

**Contributions.** In this paper, we introduce flow regularization (FlowReg), an unsupervised regularization technique for sequential Markov decision-making models that aligns the agent's latent representation field with the underlying semantic environment dynamics. It does so by learning a neural ODE that acts as a latent surrogate for the environment and aligning its flows with the latent trajectories of the agent's state embedder. To showcase our technique, we evaluate FlowReg in the reinforcement learning settings of Advantage Actor-Critic (A2C) on 11 Atari environments. Our experiments show that FlowReg notably improves the baseline model performance across all environments. We further examine the resulting latent trajectories and demonstrate their desirable smoothness properties as a result of flow-regularization. Lastly, we also show the FlowReg boost to PPO on gridworld environments.

## 2 RELATED WORK

**Neural ODEs as continuous-depth networks.** It has been noted in several existing works that ResNets (He et al., 2016) can be viewed as an Euler discretization of a continuous differential flow (Balázs et al., 2021; Lu et al., 2018; Haber & Ruthotto, 2017). An implication of this is that an ODE can, in theory, be used to model an infinite-depth ResNet with a finite number of parameters – making them more parameter efficient (Chen et al., 2018). In this paper, we take a broader view of sequence transformations modeled by the whole network as an embedder, rather than transformations modeled by the individual layers within the model. That is, instead of looking at the embedder network as a discretized transformation of an object, we look at the latent trajectories that result from applying the network to a sequence of objects that are sequentially related under well-defined environment dynamics.

**Neural ODEs for continuous control.** Neural ODEs can model the continuous evolution between discrete events while coupling with event-triggered mechanisms or classifiers to detect and handle abrupt transitions, e.g., collisions or control mode changes (Jia & Benson, 2019; Auzina et al., 2023). By integrating traditional neural networks, these models can infer both the continuous flow and the timing or conditions of discrete switches directly from data, bypassing rigid analytical formulations. The work of Alvarez et al. (2020) bears a partial resemblance to ours in that it involves training an ODE to learn entire trajectories of continuous-space environments. However, both works fundamentally differ from our approach in that our neural ODE operates on latent trajectories while theirs aim to predict semantic trajectories, which makes them rather cumbersome to apply to discrete-space

tasks since the network's output is continuous. Similar to Du et al. (2020), they use the neural ODE as the main inference model, whereas we only use the neural ODE as a decoupled regularizer.

**Shaping representations by predictive coding.** Enhancing temporal consistency across trajectories requires moving beyond static state discriminators to objectives that model long-horizon dynamics. By fusing predictive coding with contrastive learning, representations can be shaped to maximize the mutual information between past history and future outcomes, effectively smoothing the latent space against high-frequency noise (Agarwal et al., 2021; Schwarzer et al., 2020). Methods like TACO (Zheng et al., 2023) enforce a robust temporal structure in the latent space, where state transitions are predictable from their immediate predecessors, preventing the representation from drifting due to task-irrelevant environmental stochasticity. Our method enforces a stricter notion of temporal consistency by leveraging the uniqueness of ODE flows at any intermediate point, ensuring that states are predictable given *any* of their predecessors, not only the immediate ones.

## 3 PRELIMINARIES

### 3.1 MARKOV DECISION PROCESSES

We model reinforcement learning (RL) problems as *Markov decision processes* (MDPs), defined by the tuple

$$\mathcal{M} = (\mathcal{S}, \mathcal{A}, P, r, \gamma), \tag{1}$$

where $\mathcal{S}$ is the state space, $\mathcal{A}$ the action space, $P(s' \mid s, a)$ the transition kernel, $r(s, a)$ the expected immediate reward, and $\gamma \in [0, 1)$ a discount factor. An agent samples actions $a_t \in \mathcal{A}$ according to a policy $\pi(a \mid s)$, inducing a trajectory $\tau = (s_0, a_0, r_0, \ldots)$ The objective is to maximize the expected return

$$J(\pi) = \mathbb{E}_\pi \left[ \sum_{t=0}^\infty \gamma^t r(s_t, a_t) \right] \tag{2}$$

We define the following key functions:

- The state-value function: $V^\pi(s) = \mathbb{E}_\pi \left[ \sum_{t=0}^\infty \gamma^t r(s_t, a_t) \mid s_0 = s \right]$
- The action-value function: $Q^\pi(s, a) = \mathbb{E}_\pi \left[ \sum_{t=0}^\infty \gamma^t r(s_t, a_t) \mid s_0 = s, \, a_0 = a \right]$
- The advantage function: $A^\pi(s, a) = Q^\pi(s, a) - V^\pi(s)$

### 3.2 POLICY GRADIENT METHODS

Policy gradient algorithms directly optimize a parametric policy $\pi_\theta(a \mid s)$. The policy gradient theorem (Sutton et al., 1999) states:

$$\nabla_\theta J(\pi_\theta) = \mathbb{E}_{s \sim d^{\pi_\theta}, \, a \sim \pi_\theta}[\nabla_\theta \log \pi_\theta(a \mid s) \, Q^{\pi_\theta}(s, a)] \tag{3}$$

where $d^{\pi_\theta}$ denotes the stationary state distribution under $\pi_\theta$. In practice, $Q^{\pi_\theta}$ is approximated and variance is reduced by subtracting a baseline such as $V^\pi(s)$.

### 3.3 ADVANTAGE ACTOR–CRITIC (A2C)

Actor–critic methods (Mnih et al., 2016) couple a policy model (the actor) with a value function estimator (the critic). The actor updates its parameters $\theta$ via the policy gradient, while the critic learns to estimate $V^\pi(s)$ (or $Q^\pi(s, a)$) using temporal-difference learning.

The *Advantage Actor–Critic (A2C)* algorithm improves stability by using an advantage estimator. The policy gradient update is given by

$$\nabla_\theta J(\pi_\theta) \approx \mathbb{E}\left[ \nabla_\theta \log \pi_\theta(a_t \mid s_t) \, \hat{A}_t \right] \tag{4}$$

with empirical advantage

$$\hat{A}_t = r_t + \gamma V_\theta(s_{t+1}) - V_\theta(s_t) \tag{5}$$

where $V_\theta$ is the critic parameterized by $\theta$. The critic is trained by minimizing the squared error

$$\mathcal{L}_{\text{critic}}(\theta) = \mathbb{E}_{s_t \sim \pi_\theta}\left[\left(r_t + \gamma V_\theta(s_{t+1}) - V_\theta(s_t)\right)^2\right] \tag{6}$$

$$\mathcal{L}_{\text{actor}}(\theta) = -\mathbb{E}_{s_t, a_t \sim \pi_\theta}\left[\log \pi_\theta(a_t \mid s_t)\, \hat{A}_t\right] \tag{7}$$

## 3.4 Neural Ordinary Differential Equations

A Neural Ordinary Differential Equation is defined by the continuous transformation of the hidden state $h(t)$ given by the differential equation:

$$\frac{\mathrm{d}\mathbf{h}(t)}{\mathrm{d}t} = f_\phi(\mathbf{h}(t), t), \qquad \mathbf{h}(t) = \mathbf{h}(t_0) + \int_{t_0}^{t} f_\phi(\mathbf{h}(s), s)\,\mathrm{d}s \tag{8}$$

where $f$ is a neural network parameterized by $\phi$. As such, neural ODEs differs from classical deep learning in that the neural network is used to model the system dynamics (through the state derivative) at a given time instead of modeling the entire system directly. This framework can be used to model functions that evolve over time. To seamlessly integrate neural ODEs into traditional deep learning pipeline, a differentiable numeric solver (e.g., TORCHDIFFEQ (Chen et al., 2018) or DIFFRAX (Kidger, 2021)) is typically used to evaluate the latent state function at given time points. The continuous-depth nature of Neural ODEs allows adaptive computation (e.g., varying solver step sizes), offering memory efficiency and flexible trade-offs between precision and computational cost compared to fixed-depth architectures.

A key mathematical property of Neural ODEs is their invertibility and exact gradient calculation via the adjoint state, which ensures stable training even with long integration intervals. The framework inherently accommodates irregularly sampled or continuous-time data, making them suitable for tasks like time-series modeling and dynamical systems. However, their performance hinges on numerical solver choices: explicit methods (e.g., Euler) are computationally light but may struggle with stiff systems, while implicit methods (e.g., backward differentiation) enhance stability at higher computational cost. This interplay between numerical precision, stability, and efficiency underscores the importance of solver selection in practice. Additionally, Neural ODEs enable novel architectures, such as continuous normalizing flows for density estimation, by enforcing invertibility through Lipschitz constraints on $f$. By bridging deep learning with differential equations, they provide a principled framework for understanding neural networks as dynamical systems, opening avenues for interpretability and integration with scientific machine learning.

## 4 Approach

In this section, we outline the mathematical formulation of our flow regularization technique for a general target model. As illustrated in Figure 1, our setting involves three principal fields: (1) the semantic state field defined by the environment, (2) the latent observation vector field induced by the semantic state embedder on the environment, and where each point is a vector representation of the corresponding semantic state, and (3) the latent flow vector field defined by the neural ODE (i.e., flow model). Field (2) is utilized for carrying task information from Field (1) into the latent space, while Field (3) is utilized for imposing a global latent structure that underpins Field (1). The essence of our approach is that by aligning (2) and (3), we get the best of both worlds: a latent field that captures local (state-level) and global (trajectory-level) aspects of the environment.

### 4.1 Model Setup

Generally, there are two models involved in our framework, namely a target agent model $\theta$ and a flow regularizer model $\phi$. The target model comprises a state embedder network $\mathbf{h}_\theta$ that converts semantic states into their latents, and a downstream head $F_\theta$ that produces the final task-related actions. For a state trajectory $\mathbf{s} = s_0, s_1, ..., s_{N-1}$, semantic embeddings are computed as $\mathbf{H}_\theta(s) = \mathbf{h}_\theta(s_0), \mathbf{h}_\theta(s_1), ..., \mathbf{h}_\theta(s_{N-1})$, while flow embeddings are obtained by solving the initial value problem on $\mathbf{h}_\phi(0) = \mathbf{h}_\theta(s_0)$:

$$\mathbf{H}_\theta(s) = \{\mathbf{h}_\theta(s_i)\}_{i=0}^{N-1} = \mathbf{h}_\theta(\{s_i\}_{i=0}^{N-1}) \tag{9}$$

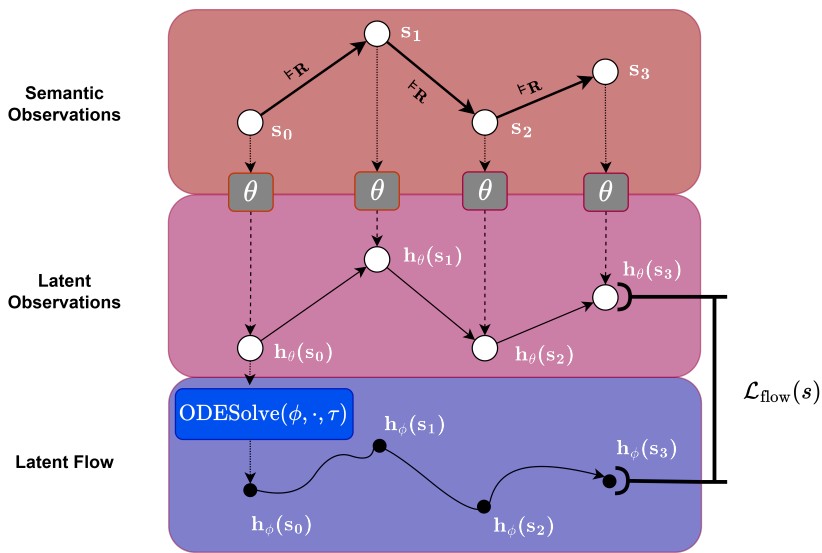

Figure 1: Illustration of the flow regularization landscape.

$$\mathbf{H}_\phi(s) = \{\mathbf{h}_\phi(s_i)\}_{i=1}^{N-1} = \text{ODESolve}(f_\phi, \mathbf{h}_\theta(s_0), \{\tau_i\}_{i=0}^{N-1}) \tag{10}$$

where $\tau_i$ is the integration time index for state $s_i$, and $f_\phi$ is a neural network that parameterizes the derivative of the latent state. MDP states generally do not have timestamps, so we impose a time sampling scheme to associate each state in the trajectory with a time index. Note that due to the Markov property, the underlying ODE is autonomous (i.e., time-invariant). However, the choice of the integration times still significantly influences the ODE solver, and our experiments show that it is indeed fairly consequential for performance. An intuitive option for time sampling would be the step index of the state, i.e., $\tau_i = i$. Another simple approach is using a discounted time horizon with the same discounting factor $\gamma$ used by the agent's algorithm, i.e., $\tau_i = \gamma^i$ where $0 < \gamma < 1$. This guarantees that integration times are in $[0, 1]$ to avoid arbitrarily large integration times, which might lead to gradient instability.

## 4.2 Path Alignment

In essence, the flow model defines a smooth latent path that starts at a given semantic state embedding point, whereas the semantic embedder defines a discrete point sequence in the latent space. Typically, this latent point sequence is topologically unconstrained, which means that the topological structure of the latent space has to be implicitly learned over the course of the training. The key idea here is that we can speed up this process by imposing a topological structure that we already know to be compatible with the domain.

Our approach proceeds from the rationale that initially, the flow model carries pure curvature information while the semantic embedder carries task information. Ideally, we want to fuse both signals into the target model. To that end, we align the semantic embedding trajectory with the discretized latent flow. In doing so, each network adapts the information carried by the other. One straightforward way to incentivize this alignment is by minimizing the MSE between the latent point sequence $\mathbf{H}_\theta$ and the sampled flow path $\mathbf{H}_\phi$. As such, we can compute the flow regularization loss as follows:

$$\mathcal{L}_{\text{flow}}(s) := \frac{\|\mathbf{H}_\theta(s) - \mathbf{H}_\phi(s)\|_2^2}{N} \qquad \text{(FlowReg)} \tag{11}$$

### 4.3 Overall Training Objective

Having computed the flow loss on the latent trajectory, this loss is then added to the label-based task loss:

$$\mathcal{L}(s, y) = \mathcal{L}_{\text{task}}(F_\theta(\mathbf{H}_\theta(s)), y) + \lambda\mathcal{L}_{\text{flow}}(s) \tag{12}$$

where $\lambda$ is the flow-loss weighting factor. Note that $\mathcal{L}_{\text{flow}}(s)$ involves both the semantic embedder $\theta$ and the neural ODE network $\phi$. This trains $\theta$ to follow the continuous ODE flow while optimizing $\phi$ to indirectly adapt to the underlying task modeled by $\theta$.

For an Advantage Actor-Critic agent, the overall training loss would be:

$$\mathcal{L}(s, y) = \mathcal{L}_{\text{actor}}(s, y) + \beta\mathcal{L}_{\text{critic}}(s, y) + \lambda\mathcal{L}_{\text{flow}}(s) \tag{13}$$

A relevant hyperparameter here is the FlowReg update frequency relative to the agent policy updates. It is also important to note that the neural ODE is not used for inference, only as a training-time adaptive regularizer.

## 5 Experiments

We evaluate our method on 11 Atari environments from the Arcade Learning Environment (ALE) library (Bellemare et al., 2013). This is mainly due to A2C being a reasonably simple actor-critic formulation, which is a cornerstone for many state-of-the-art algorithms like PPO (Schulman et al., 2017) and SAC (Haarnoja et al., 2018). We build on the Stable-baselines3 A2C implementation (Raffin et al., 2021) to incorporate our regularization loss. We use the same set of A2C hyperparameters for all environments and agents. The agent networks for both baseline and flow-regularized variants are identical for all experiments. The ultimate goal of our evaluation is to show that flow regularization effectively reduces the training search space by imposing an ODE flow field on the latent space of the agent's state embedder, hence greatly reducing variance during training, allowing the agent to learn better policies with the same training steps.

### 5.1 Atari Benchmarks

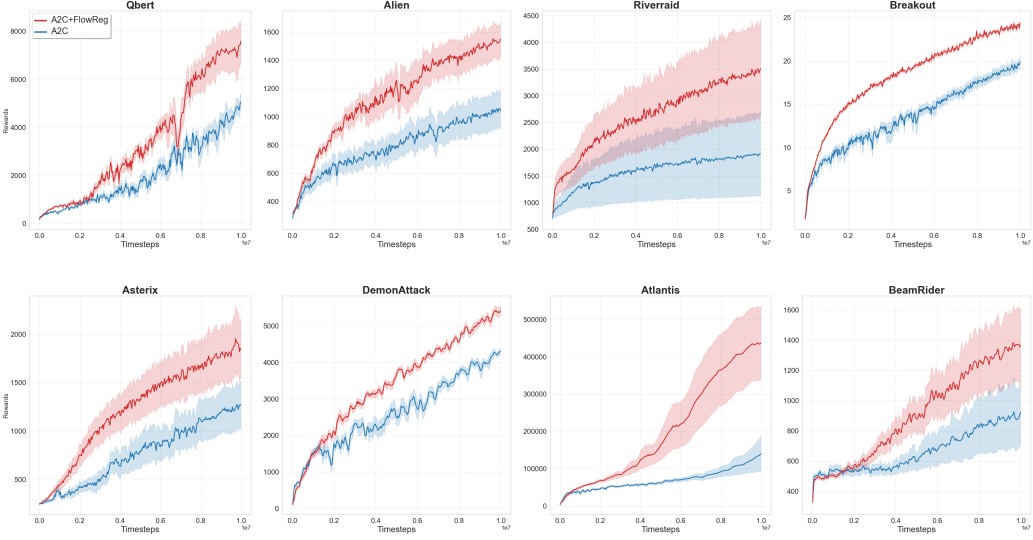

Figure 2: Episodic rewards of baseline and flow-regularized A2C on 8 different Atari environments with a rolling average window of 100 episodes.

**Hyperparameters.** We performed 5 independent runs for every RL agent across all environments for 10 million timesteps each. Our semantic embedder for both baseline and flow-regularized agents is a commonly used Nature CNN (Mnih et al., 2015) feature extractor that embeds game state (frames) into a 512-dimensional vector space. The ODE flow (and loss) is computed on the extracted state feature vectors. For the FlowReg ODE network, we use a two-layer MLP with a `tanh` activation on the first layer. All models are optimized by RMSProp (Ruder, 2016) with an initial learning rate of $7 \times 10^{-4}$ and a linear decay scheduler. We apply a global-norm gradient clipping ratio of 0.5 (Pascanu et al., 2012). We use the TORCHDIFFEQ (Chen et al., 2018) library together with PY-TORCH for solving neural ODEs with relative tolerance $=10^{-4}$, and absolute tolerance $=10^{-5}$. For FlowReg variants, we experiment with both index-based ($\tau_i = i$) and exponential decay ($\tau_i = \gamma^i$) time sampling, along with a regularization frequency (relative to agent updates) of $\{5, 10, 20\}$, and take the best configuration averaged over 3 seeds dedicated for hyperparameter search and separate from the 10 seeds of the final comparison runs. For simplicity, we set $\lambda = 1$ for all environments.

**Flow-regularized agents consistently outperform the baseline on Atari environments.** Figure 2 highlights the notable performance gap between flow-regularized A2C and the baseline. The learning curves on all 11 environments can be found in Figure 5 (Appendix A). Figure 3 shows the overall performance percent gains achieved by applying FlowReg on all 11 environments[1]. We also find that most FlowReg configurations outperform the baseline across all environments, which means that finding good values for the two FlowReg hyperparameters (time sampling and update frequency) is fairly easy.

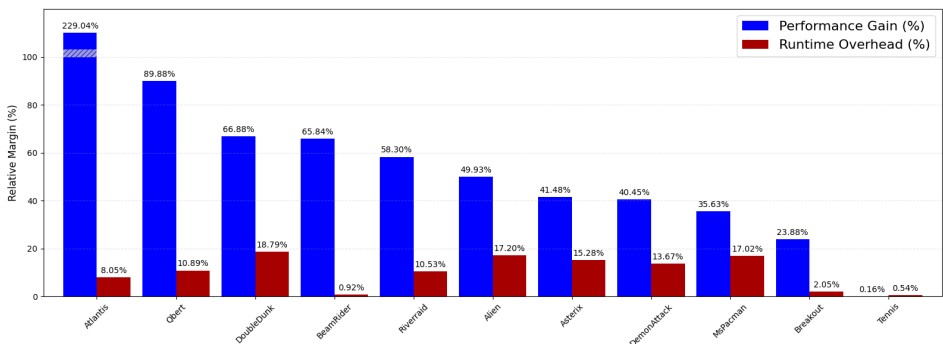

Figure 3: Trade-off between performance gain achieved by FlowReg and its runtime overhead.

**FlowReg performance gains are robust under time sampling modes.** As shown in Table 1, FlowReg largely improves the baseline performance under both *Index* and *Exp-Decay* time sampling modes. The choice between them, in all likelihood, depends on the granularity of the environment dynamics. We generally expect *Exp-Decay* to work better on environments with swifter or more fine-grained state transitions. Table 4 (Appendix B) shows the specific FlowReg configurations that performed best on each environment along with the corresponding runtimes.

Table 1: Best mean episode rewards of different time sampling modes. Each variant was evaluated on 16 episodes averaged across 10 different training seeds. *Index* is where $\tau_i = i$ and *Exp-Decay* is where $\tau_i = \gamma^i$.

| A2C AGENT | QBERT | RIVERRAID | BEAMRIDER |
|---|---|---|---|
| BASE | $4374.30 \pm 958.42$ | $1862.27 \pm 2399.58$ | $960.66 \pm 748.23$ |
| FLOWREG (INDEX) | $\mathbf{8306.05 \pm 1752.71}$ | $2946.34 \pm 2788.17$ | $1590.96 \pm 1033.30$ |
| FLOWREG (EXP-DECAY) | $6903.15 \pm 2157.71$ | $\mathbf{2947.95 \pm 2798.64}$ | $\mathbf{1593.11 \pm 961.77}$ |

---

[1]The hatched strip in Figure 3 indicates values exceeding the y-axis limit, which was capped for visual clarity to avoid overly downscaling other values.

Table 2: Mean episode rewards of different FlowReg update frequencies relative to agent updates on Atari Qbert. Each variant was evaluated on 16 episodes averaged across 10 different training seeds. **U-m** means the FlowReg loss is applied once every **m** agent updates.

| A2C AGENT | QBERT (INDEX) | QBERT (EXP-DECAY) |
|---|---|---|
| BASE | $4374.30 \pm 958.42$ | $4374.30 \pm 958.42$ |
| FLOWREG U-5 | $\mathbf{8306.05 \pm 1752.71}$ | $5286.60 \pm 1269.76$ |
| FLOWREG U-10 | $6569.51 \pm 2645.71$ | $\mathbf{6903.15 \pm 2157.71}$ |
| FLOWREG U-20 | $5985.70 \pm 2756.17$ | $6782.70 \pm 1877.13$ |

**FlowReg loss is still effective under a much lower update frequency compared to the agent loss.** Table 2 points to it being more ideal to apply FlowReg loss once every 10 agent updates under both time sampling modes. The fourth row (U-20) also shows that FlowReg still results in notable performance gains with half as many updates. This is good news for runtime as it means the FlowReg loss does not need to be aggressively optimized to improve over the baseline, which allows it to run in a comparable training time. By contrasting the time-overhead margins with the performance gains in Figure 3, it shows that FlowReg is an overall cost-effective choice. Figure 6 and Table 4 (Appendix B) show the runtime comparison between the baseline and FlowReg in terms of absolute values.

Table 3: Latent path smoothness measures normalized by trajectory length.

| ENV | METRIC
FORMULA | PATH LENGTH
$\sum_{t=0}^{N-1}\|\Delta\mathbf{h}_\theta(\mathbf{s_t})\|$ | NET DISPLACEMENT
$\|\mathbf{h}_\theta(\mathbf{s_{N-1}}) - \mathbf{h}_\theta(\mathbf{s_0})\|$ | ACCEL. ENERGY
$\sum_{t=0}^{N-2}\|\Delta^2\mathbf{h}_\theta(\mathbf{s_t})\|$ | REWARD
$\sum_{t=0}^{N} R_t$ |
|---|---|---|---|---|---|
| QBERT | A2C | $34.39 \pm 2.14$ | $0.44 \pm 0.17$ | $4424.75 \pm 521.76$ | $4374.30 \pm 958.42$ |
| | A2C+TACO | $6.13 \pm 0.42$ | $\mathbf{0.03 \pm 0.01}$ | $106.38 \pm 9.86$ | $2434.05 \pm 2474.44$ |
| | **A2C+FLOWREG** | $\mathbf{4.20 \pm 0.44}$ | $0.10 \pm 0.02$ | $\mathbf{64.17 \pm 7.05}$ | $\mathbf{8306.05 \pm 1752.71}$ |
| BREAKOUT | A2C | $104.09 \pm 2.44$ | $0.74 \pm 0.28$ | $31432.59 \pm 1698.82$ | $19.40 \pm 1.86$ |
| | A2C+TACO | $13.09 \pm 1.08$ | $0.13 \pm 0.05$ | $461.75 \pm 125.72$ | $11.12 \pm 2.42$ |
| | **A2C+FLOWREG** | $\mathbf{4.92 \pm 0.23}$ | $\mathbf{0.06 \pm 0.02}$ | $\mathbf{94.98 \pm 9.51}$ | $\mathbf{24.03 \pm 0.84}$ |
| RIVERRAID | A2C | $75.36 \pm 2.72$ | $0.53 \pm 0.07$ | $18298.55 \pm 1487.28$ | $1862.27 \pm 2399.58$ |
| | A2C+TACO | $50.35 \pm 1.26$ | $0.36 \pm 0.04$ | $7599.32 \pm 404.50$ | $2943.47 \pm 1616.30$ |
| | **A2C+FLOWREG** | $\mathbf{6.35 \pm 0.29}$ | $\mathbf{0.06 \pm 0.02}$ | $\mathbf{137.25 \pm 10.11}$ | $\mathbf{2947.95 \pm 2798.64}$ |

## 5.2 LATENT PATH SMOOTHNESS

In addition to the performance results, we set out to investigate some geometric properties of the latent paths (trajectories) of flow-regularized models compared to the baseline. In particular, we are interested in whether FlowReg induces smoother paths as a result of the ODE alignment. We measure 3 different smoothness metrics as shown in Table 3. All 3 metrics are computed on the full dimensionality of the latent space without any reduction, and $\|\cdot\|$ is the Euclidean norm. To control for trajectory length variations, all 3 metrics are normalized by trajectory length, so they correspond to average speed, velocity, and acceleration, respectively.

Path length measures total segment length along the path, which reflects the jump step size between consecutive states in the latent space. Ideally, latent representations of consecutive states should be in close proximity, so the smaller the path length, the better the state embedder is from a purely topological standpoint. Lower net path displacement is desirable for similar reasons, as it indicates that individual trajectories lie in tightly packed regions of the latent space. Acceleration energy, computed the second-difference in position: $\Delta^2\mathbf{h}_\theta(s_i) = \mathbf{h}_\theta(s_{i+2}) - 2\mathbf{h}_\theta(s_{i+1}) + \mathbf{h}_\theta(s_i)$, is a more local measure roughness (lower is better).

**FlowReg results in much smoother latent trajectories while improving overall performance.** Table 3 shows that ODE flow alignment notably changes the basic geometric properties of the agent's latent trajectories, making them much smoother and more tightly wound, consistently across environments. Naturally, we do not attribute the performance improvement solely to the latent trajectory

smoothing effect, since there are many ways to smooth the space while destroying the semantic structure, as evident by the fact that although TACO produces smoother paths than baseline over all 3 environments, it leads to a considerable performance degradation on two of them. The key distinction in this case is restricting the latent field while respecting the underlying transition dynamics. In our case, this is achieved by the mutual alignment loss that imposes a diffeomorphic structure on the latent space, resulting in reduced variance as abrupt jumps and crossings are naturally penalized because they violate ODE flows.

Another takeaway from Table 3 is that smoothness and temporal predictability are notably correlated. Despite the differences in mechanism between TACO and FlowReg, they both aim to instate a notion of predictive temporal structure on the latent representations. The results of Table 3 suggest that this common feature explains the notable reduction in their latent path roughness compared to the baseline.

## 5.3 MINIGRID ENVIRONMENTS

We evaluate FlowReg on PPO (Schulman et al., 2017) in Minigrid environments (Chevalier-Boisvert et al., 2023). These experiments serve the purposes of showing FlowReg's efficacy on another major RL algorithm (PPO) while also exploring a more radically discrete environment domain than Atari games. Similar to A2C, we use a modified implementation of the Stable-Baselines-3 PPO (Dhariwal et al., 2017). We use the *Index* U-20 FlowReg configuration for all 3 environments. We performed 10 runs per agent for 1M timesteps each.

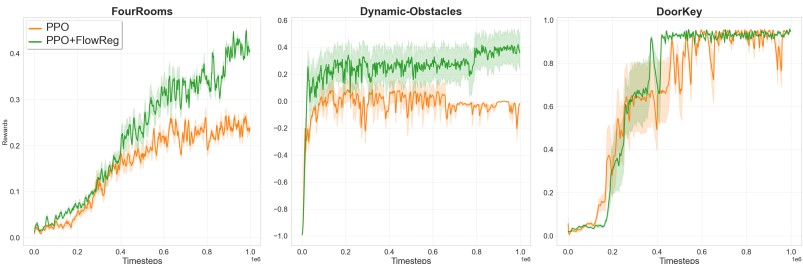

Figure 4: Episodic rewards of baseline and flow-regularized PPO on Minigrid environments with a rolling average window of 100 episodes.

As shown in Figure 4, flow-regularized PPO has a clear advantage on *FourRooms* and *Dynamic-Obstacles* while matching the baseline in *DoorKey*, where both agents practically solve the environment.

## 6 CONCLUSION

**Summary.** In this paper, we presented FlowReg, an unsupervised regularization technique that aligns MDP semantic trajectories with their latent counterparts. We realized this goal by adding an unsupervised loss term that incentivizes the semantic trajectory embeddings to act like discretizations of a global neural ODE flow. We chose actor-critic reinforcement learning on Atari and Minigrid environments to showcase the benefits of applying FlowReg to a target model. Our results have shown that using FlowReg notably boosts the overall performance of the target agent across almost all attempted environments and results in a more constrained path structure on the learned embedding space.

**Limitations.** Although FlowReg does not require full episodes, it still requires trajectory information to align it with the learned ODE flow. This means the training pipeline needs to keep track of the episode ID for each state-action pair. This is not a significant challenge for the classical RL pipeline structure, where each batch resumes from the environment state after the previous batch. However, this might impose more implementation demands on more complex pipelines that do not place as much emphasis on episodic structure. A more fundamental limitation of FlowReg is the fact

that ODE flows are unique both forwards and backwards, so flow paths do not intersect themselves or each other. This can be beneficial for discouraging looping behavior where an agent returns to a previously visited state. However, this property could present a burden in environments where there are intermediate bottleneck states that need to be passed from different starting states. An example of that is a maze solver game where the target destination lies in a chamber with only one opening. Fortunately, this is often not the case for environments with a very large state space (like Atari).

**Future Work.** Since experiments demonstrate the efficacy of FlowReg on a standard on-policy RL algorithm, it would be of great interest to see how it fares in the off-policy settings such as DQN (Mnih et al., 2013), as well as model-based algorithms like Dreamer (Okada & Taniguchi, 2021). Although the scope of our evaluation pertains to RL, the method itself still lends itself to MDPs in other learning paradigms such as imitation learning or semi-supervised learning. As such, these investigations would be a very promising research direction.

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

# A APPENDIX

## A.1 LEARNING CURVES ON ALL ENVIRONMENTS

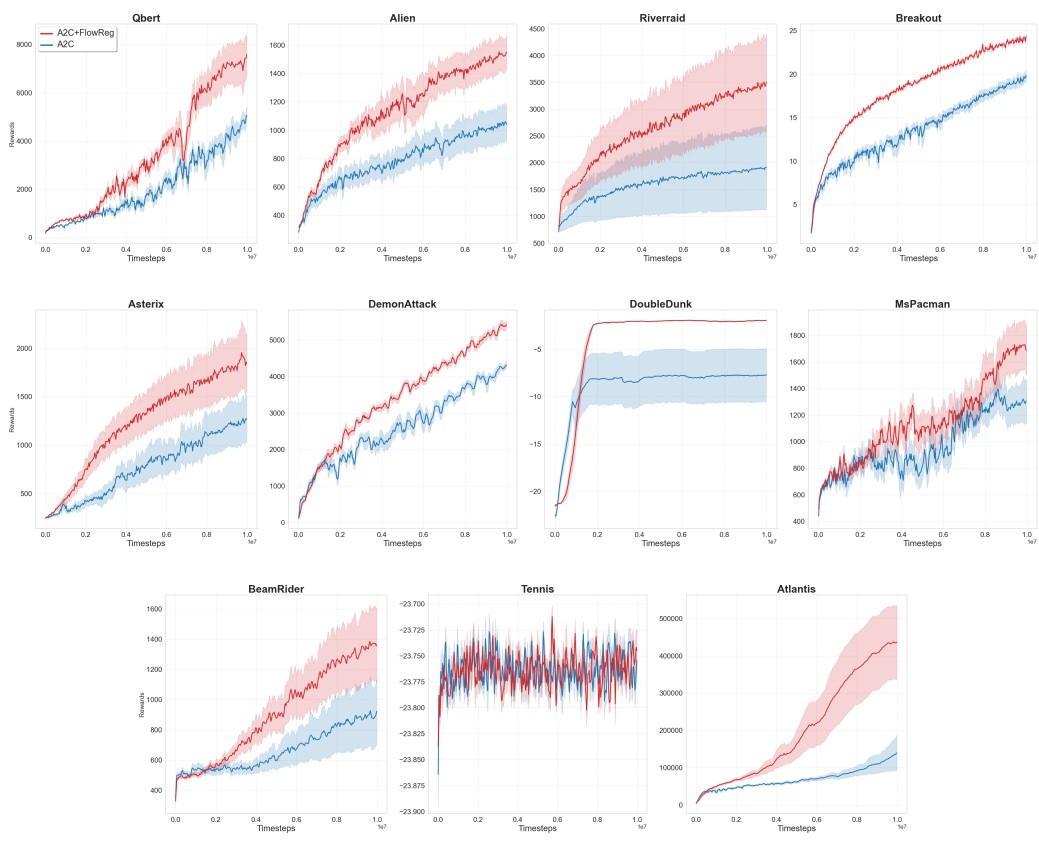

Figure 5: Episodic rewards of baseline and flow-regularized A2C on all 11 Atari environments with a rolling average window of 100 episodes.

# B FLOWREG CONFIGURATIONS AND RUNTIME

Table 4: FlowReg configurations used for each environment and their corresponding runtimes.

| ENVIRONMENT | TIME SAMPLING | REL. UPDATE FREQUENCY | A2C RUNTIME (MIN.) | A2C+FLOWREG RUNTIME (MIN.) | RUNTIME OVERHEAD (%) |
|---|---|---|---|---|---|
| DEMONATTACK | EXP-DECAY | 10 | 487.37 | 554.00 | 13.67 |
| ATLANTIS | EXP-DECAY | 10 | 603.44 | 652.00 | 8.05 |
| BEAMRIDER | EXP-DECAY | 20 | 561.82 | 567.00 | 0.92 |
| TENNIS | EXP-DECAY | 20 | 617.68 | 621.00 | 0.54 |
| RIVERRAID | EXP-DECAY | 5 | 632.40 | 699.00 | 10.53 |
| ASTERIX | EXP-DECAY | 5 | 414.66 | 478.02 | 15.28 |
| MSPACMAN | EXP-DECAY | 5 | 538.13 | 629.70 | 17.02 |
| QBERT | INDEX | 5 | 510.13 | 565.70 | 10.89 |
| BREAKOUT | INDEX | 5 | 775.07 | 791.00 | 2.05 |
| DOUBLEDUNK | INDEX | 5 | 1011.86 | 1202.00 | 18.79 |
| ALIEN | INDEX | 5 | 691.99 | 811.00 | 17.20 |

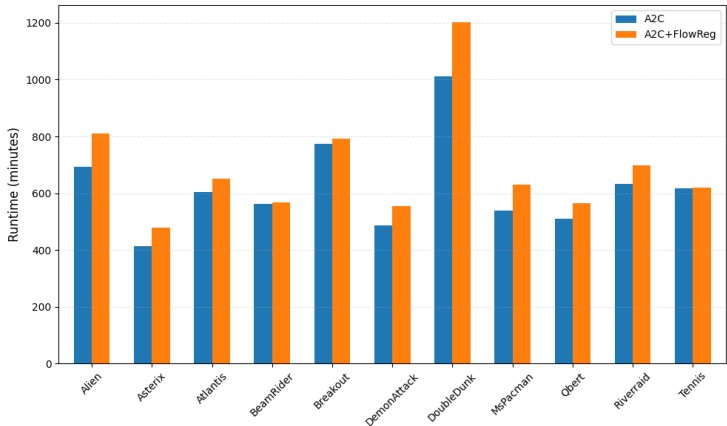

Figure 6: Total Training Runtime Comparison (for 10M timesteps).

## C HYPERPARAMETER TUNING EXPERIMENTS

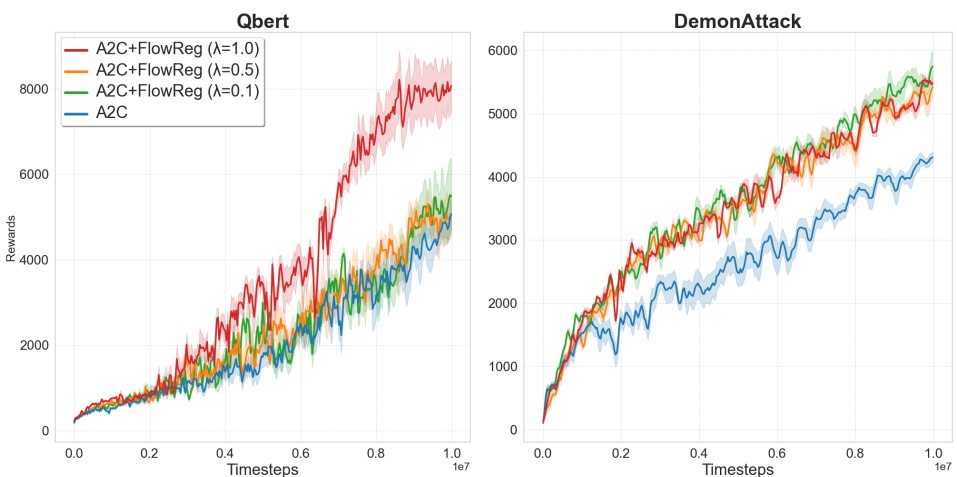

Figure 7: Performance of different FlowReg loss weights ($\lambda$).

