# OpenReview forum: "Flowing Through States: Neural ODE Regularization for Reinforcement Learning"
_ICLR.cc/2026/Conference — ICLR 2026 Poster_

### Official Review · Reviewer_KaXh · 2025-10-24

**Soundness:** 3
**Presentation:** 2
**Contribution:** 1
**Rating:** 4
**Confidence:** 4

**Summary:**

The work proposes a new regularization technique to facilitate learning of useful representations for reinforcement learning based on neural ODEs. Motivated by trajectories within MDPs to follow the dynamics of the transition function, the authors argue that latent representations learned by the agent should try to capture such dynamics. In addition to a encoder network that encodes states into a latent space, but might not capture the structure of the MDP transitions, they propose to learn a neural ODE model that captures the flow of latent embeddings throughout time. To encourage the latent space to follow the flow of the ODE model, they introduce an additional regularization loss term that minimizes the difference between the embeddings obtained by both models. The proposed FlowReg approach is evaluated on top of A2C in 10 Atari game environments and shown to improve sample efficiency. Additional analyzes shows the effect of different components of the ODE model, including time sampling approaches, update frequency, and shows that the latent space is smoother compared to without the regularization.

**Strengths:**

The discussed problem of learning representations for sequential decision making that smoothly capture the dynamics of the environment is highly relevant and important. To the best of my knowledge, the application of neural ODEs to learn and regularize the flow of latent states is novel and original. The method is largely clearly defined in Section 4 and is conceptually fairly simple which I appreciate.

Experiments in 10 Atari environments show that FlowReg consistently leads to improved performance with gains varying between tasks. There appears to be no task in which FlowReg harms performance which is encouraging. Additional ablations/ experiments demonstrate the robustness of the approach to different configurations and quantitatively analyzes the learned latent space in Table 3.

Overall, I would consider this paper a largely well executed work, albeit with potentially limited significance and a lack of contextualization within related literature, as per weaknesses below.

**Weaknesses:**

In its current form, I am afraid that the work is not sufficient to justify acceptance at a venue like ICLR. Below, I try to give concrete weaknesses and highlight any weaknesses that I see as critical / major with (**Major**). I'd expect these to be addressed for this work to be considered for acceptance.

## Originality & Significance
As mentioned above, I consider the combination of neural ODEs to model flow of latent states in MDPs as a regularization technique original and interesting. However, I am not convinced that it is a significant contribution to the field.

In its current form, this work does not at all acknowledge, discuss, or compare to alternative approaches of shaping representations for reinforcement learning. This is a rich literature space that I would highly suggest the authors to review and discuss in detail within their work in order to establish any potential unique features that their approach might offer. Almost all of these approaches consider the structure of a MDP in some form (temporal proximity, transition dynamics, ...) to shape representations that facilitate more efficient learning. The following might provide some starting points for such a review but is really just the tip of the iceberg:
- Approaches that learn representations via constrastive learning
	- Uses contrastive learning to learn representations in which observations and noise-injected / augmented observations are close in latent space: Laskin, Michael, Aravind Srinivas, and Pieter Abbeel. "Curl: Contrastive unsupervised representations for reinforcement learning." In _International conference on machine learning_, pp. 5639-5650. PMLR, 2020.
	- Use contrastive learning to learn similar representations for states and actions that are close in time: Zheng, Ruijie, Xiyao Wang, Yanchao Sun, Shuang Ma, Jieyu Zhao, Huazhe Xu, Hal Daumé III, and Furong Huang. "TACO: Temporal Latent Action-Driven Contrastive Loss for Visual Reinforcement Learning." _Advances in Neural Information Processing Systems_ 36 (2023): 48203-48225.
- Approaches that learn representations leveraging temporal structure of transitions via predicting the future.
	- Schwarzer, Max, Ankesh Anand, Rishab Goel, R. Devon Hjelm, Aaron Courville, and Philip Bachman. "Data-efficient reinforcement learning with self-predictive representations." In _International conference on learning representations_, 2021.
	- McInroe, Trevor, Lukas Schäfer, and Stefano V. Albrecht. "Multi-horizon representations with hierarchical forward models for reinforcement learning." In _Transactions on Machine Learning Research_, 2024.
- Approaches that learn similar representations for states that share certain properties according to bisimulation metrics
	- Agarwal, Rishabh, Marlos C. Machado, Pablo Samuel Castro, and Marc G. Bellemare. "Contrastive behavioral similarity embeddings for generalization in reinforcement learning." In _International conference on learning representations_, 2021.
	- Castro, Pablo Samuel, Tyler Kastner, Prakash Panangaden, and Mark Rowland. "MICo: Improved representations via sampling-based state similarity for Markov decision processes." _Advances in Neural Information Processing Systems_ 34 (2021): 30113-30126.
- Approaches that leverage inverse dynamics models to learn representations for decision making
	- Pathak, Deepak, Pulkit Agrawal, Alexei A. Efros, and Trevor Darrell. "Curiosity-driven exploration by self-supervised prediction." In _International conference on machine learning_, pp. 2778-2787. PMLR, 2017.
- ...

There are three main questions/ expectations within this that I'd all consider **major** weaknesses that need addressing:
1. Given all these works in some form are motivated by the same question as this work -- how can we leverage the structure of the sequential problem of a MDP to learn better representations for decision making -- I would expect a detailed discussion that makes any differences in learned representations or assumptions clear between this work and prior work.
2. Does FlowReg have any unique benefits over these alternatives?
3. To establish any significance of this work in the context of the prior literature, I would expect comparisons of the effects and benefits of FlowReg with alternative recent representation learning approaches. And just to be clear, I am not stating that this method needs to be state-of-the-art on some benchmark to warrant publication, but given the similarity in objectives, there should be an empirical comparison to better understand potential differences.

## Methodology
4. **Major:** The proposed ODE model defined in Eq. (9, 10) appears to be purely state and time conditioned. That is somewhat confusing to me as transitions to future states depend on the actions being taken by the agent. In that sense, the function that the neural ODE is trying to learn changes whenever the policy changes which is constantly happening throughout training. Is my understanding correct that the objective of the neural ODE is non-stationary as it changes with the current policy generating trajectories? If so, do you find this to be a problem for the learning process of the ODE? What does the ODE tend to learn?

## Experiments
5. **Major**: To further substantiate weakness 3. above, I believe that comparisons to existing auxiliary representation learning objectives would be valuable. The main benefits and claims of FlowReg appears to be (1) improved performance/ learning efficiency and (2) smoothness of the latent space. However, approaches such as temporal contrastive auxiliary losses or future-predictive auxiliary objectives tend to also learn a latent space in which embeddings that are close in time are also close in embedding space, resulting in a latent states that evolve smoothly in time. Is FlowReg different in its learned latent space from these alternatives? To answer this question, I would expect to see some results for recent representatives of such alternative approaches that compares their latent space to FlowReg either qualitatively using visualisations or quantitatively using metrics as presented in Table 3.
6. **Major**: The metrics presented in Table 3 to evaluate the smoothness of the latent space appear to be potentially uninformative. While a smaller path length and displacement might indicate a smoothly evolving latent space throughout the trajectory, it is also possible that the learned latent space merely compressed embeddings into a smaller space while maintaining identical smoothness and structure. For example, one could scale a latent space with an arbitrary small positive factor and obtain a new latent space that exhibits significantly lower path length and displacement. However, I would argue this latent space would be no smoother than the unscaled previous latent space. Such shortcuts can easily be learned through the use of regularization techniques as Eq. (12) merely encourages to minimize the distance of embeddings. That could likely be achieved by scaling down the latent space. To convincingly show the smoothness of the latent space, it would be helpful to visualize PCA/ t-SNE/ UMAP projections of the trajectories within latent space and see whether they are indeed more smooth compared to baselines.
7. The novel contribution of this work is its ODE regularization process. Connected to questions in Weakness 4. Is there any qualitative analysis that you could provide that sheds insight into what the ODE tends to learn?
8. Figure 3 states to show "overall relative performance gains" but it is unclear what exactly is being reported. Would the authors be able to clarify what they compute for this Figure?
9. What do lines and shading in Figure 2 correspond to? I would have typically expected them to correspond to mean and standard deviation or confidence intervals but given the noisy lines with small shading for example in DemonAttack and Qbert, I would doubt that is the case.
10. As per Figure 3, relative performance gains of FlowReg differ quite substantially across environments. Do you have any insight into what makes FlowReg particularly effective in BeamRider and Qbert but less effective in Tennis? Are there any particular properties within the tasks with more significant gains that allow for the ODE model to be more precise?
11. The authors state that they use the same hyperparameters for all environments and agents (Section 5). How were these hyperparameters determined?

## Clarity
12. The work frequently uses the terms of "semantic space", "semantic trajectories", and "semantic observations" in contrast to latent space and information. However, it is not clear to me what exactly these terms are meant to express. It would be helpful if these terms are properly introduced and/ or defined.
13. Within the introduction, the authors state that their method "combines the expressivity of continuous-time dynamics with the efficiency of conventional neural architectures" but it is unclear to me why modelling continuous-time dynamics within MDPs is valuable given the MDP itself operates on discrete steps.
14. Within the introduction, the authors states "the structure properties it [embedding function] imposes on the latent space -- such as smoothness, consistency, and determinism -- are far from trivial and are crucial for reasoning tasks." It is unclear to me why it is crucial for the embedding function to impose determinism. Do the authors refer to the encoding from states to latent embeddings to be deterministic, or the transitions between states to be deterministic within the latent space? Any clarification and substantiation of this claim would be appreciated.
15. Function $g$ in the unnumbered equation within the introduction appears undefined. What does $g$ correspond to?
16. Most parameterised functions are written with parameters as a subscript but in Eq. (8), you write $f(..., \phi)$ instead of $f_\phi(...)$. Also, the right hand equation of Eq. (8) also write $f$ but without any notation to indicate its parameters.

**Questions:**

1. What do you refer to with the term "semantic space" and how is it different from the latent space obtained from any encoder network? (Weakness 12)
2. Function $g$ in the unnumbered equation within the introduction appears undefined. What does $g$ correspond to? (Weakness 15)
3. How does FlowReg differ from existing auxiliary objectives to shape representations for sequential decision making (discussion of related literature above)? Does FlowReg have any unique advantages? (Weaknesses 1/2)
4. The neural ODE process appears to be just conditioned on states and time. Is my understanding correct that the learning objective of this process is non-stationary since the flow of states throughout time depends on the policy -- that is learned and continually changes -- used to collect these trajectories? If so, do you find this to be a problem for the learning process of the ODE? (Weakness 4)
5. Would you be able to show some qualitative analyzes and visualizations of the latent space learned by the ODE model and the latent space resulting from FlowReg? This might help to better understand what is being learned, and how it might be beneficial. (Weaknesses 5/6)
6. How are the "overall relative performance gains" values reported in Figure 3 computed? (Weakness 8)
7. What do lines and shading in Figure 2 correspond to? (Weakness 9)
8. Do you have any insight into what makes FlowReg particularly effective in BeamRider and Qbert but less effective in Tennis? Are there any particular properties within the tasks with more significant gains that allow for the ODE model to be more precise? (Weakness 10)
9. The authors state that they use the same hyperparameters for all environments and agents (Section 5). How were these hyperparameters determined? (Weakness 11)

---

> ### Author Response · Authors · 2025-11-22
>
> > What do lines and shading in Figure 2 correspond to?
>
> - They correspond to the mean and standard deviation of the reward (the y-axis). This is a standard learning curve.
>
> > What do you refer to with the term "semantic space" and how is it different from the latent space obtained from any encoder network?
>
> - The semantic space simply refers to the space of environment states. In the case of Atari games, the semantic state is the 84x84x3 frame (image) at a given step, so the semantic space $\subseteq \mathbb{R}^{84\times84\times 3}$. The latent space contains the representations/embeddings of the semantic states. For instance, if the encoder network is a feature extractor that outputs 512 features, then the latent space $\subseteq \mathbb{R}^{512}$.
>
> > For example, one could scale a latent space with an arbitrary small positive factor and obtain a new latent space that exhibits significantly lower path length and displacement.
>
> - There are, of course, many ways to artificially cheat these metrics by external intervention (the same goes for t-SNE, by the way) without achieving anything useful. The point of our experiment, however, is that this smoothing happens organically as a byproduct of flow-regularization while also greatly improving the underlying performance.
>
> > Function $g$ in the unnumbered equation within the introduction appears undefined. What does $g$ correspond to?
>
> - $g$ is any arbitrary function. The equation is meant to express that ideally, the embedding of the next state $h(s_2)$ should purely be a function of the current state embedding $h(s_1)$ and the transition rules $R$.
>
> > How does FlowReg differ from existing auxiliary objectives to shape representations for sequential decision making (discussion of related literature above)? Does FlowReg have any unique advantages?
>
> - The literature you linked is rather broad, so we will try to address its general relevance concisely in the following:
> - Contrastive learning (CL) tackles the problem from a different angle than ours. CL generally works with vicinities (related points are grouped nearby) while ODE flows mark paths that can relate points that are spatially very remote. As such, the connection between CL and our approach is rather weak.
> - Approaches that rely on semantic similarity between states are also quite distinct from ours. We do not rely on a domain-specific notion of semantic similarity.
> - A more viable argument can be made for self-predictive methods since our ODE flow is essentially trained to predict the latent trajectories induced by the policy. In that sense, our work can be seen in loose connection to curiosity-driven approaches, except that they typically reshape the reward while we operate on state representations. Therefore, we shall reference them in our Related Work.
>
> > Is my understanding correct that the learning objective of this process is non-stationary since the flow of states throughout time depends on the policy -- that is learned and continually changes -- used to collect these trajectories? If so, do you find this to be a problem for the learning process of the ODE?
>
> - Yes, this is a non-stationary objective (by design). This is very common in RL settings. For example, the A2C objective is also non-stationary since both the actor and critic are constantly updated. Our results are evidence that this is not a critical problem for the learning process, given that the flow-regularized agents largely outperform the baseline.
>
> > Would you be able to show some qualitative analyzes and visualizations of the latent space learned by the ODE model and the latent space resulting from FlowReg?
>
> - Visualizing a highly dimensional latent space (512 dimensions) while preserving the spatial/topological structure is rather difficult (if possible at all). We experimented with t-SNE and Isomap, but they both disfigured the geometric properties of the trajectories/paths.
>
> > How are the "overall relative performance gains" values reported in Figure 3 computed?
>
> - The performance gains are computed as: $\frac{R_{FlowReg} - R_{Base}}{R_{Base}}*100\\%$ where $R$ is the mean evaluation reward.
>
> > Do you have any insight into what makes FlowReg particularly effective in BeamRider and Qbert but less effective in Tennis?
>
> - Given that the baseline also fails to learn on Tennis, it might be an issue with the common A2C hyperparam configuration, but we haven't investigated the environment in depth.
>
> > The authors state that they use the same hyperparameters for all environments and agents (Section 5). How were these hyperparameters determined?
>
> - They base A2C hyperparameters (shared by all experiments) were a combination of traditional SB3 settings and our own tuning.
>
> > Most parameterised functions are written with parameters as a subscript but in Eq. (8)..
>
> - We unified the parameter subscript convention for Eq. (8) in the revision.

---

> > ### Comment · Reviewer_KaXh · 2025-11-24
> > **Rebuttal response**
> >
> > I thank the authors for their comments and addressing many of my questions.
> >
> > > The semantic space simply refers to the space of environment states.
> >
> > In that case, I'd suggest to refer to the space as state space which is a standard term in contrast to semantic space.
> >
> > > There are, of course, many ways to artificially cheat these metrics by external intervention (the same goes for t-SNE, by the way) without achieving anything useful. The point of our experiment, however, is that this smoothing happens organically as a byproduct of flow-regularization while also greatly improving the underlying performance.
> >
> > I agree that visualization approaches like t-SNE/ UMAP also have their flaws, but I am not convinced that the suggested metrics show that the proposed regularization technique results in a smoother latent space given my concerns. The regularization objective could meaningfully be minimized by significantly scaling down the latent space with a small constant which would result in similar metrics. Unless the authors provide more reliable evidence for the claim that the regularization approach results in a temporally smooth latent space, I would not consider this claim to be supported with sufficient evidence.
> >
> > > Contrastive learning (CL) tackles the problem from a different angle than ours. CL generally works with vicinities (related points are grouped nearby) while ODE flows mark paths that can relate points that are spatially very remote. As such, the connection between CL and our approach is rather weak.
> >
> > While I agree that the perspective of contrastive learning is different from the proposed approach, I am not convinced that the proposed FlowReg regularization technique learns a meaningfully different latent space. Temporal contrastive objectives such as the TACO work referenced in my review, or [1, 2], aim to learn a latent space in which temporally close states are also close in latent space or at least predictive of each other. These objectives tend to result in a temporally smooth latent space in which two states that are close in time are also close in latent space. Such a latent space would obtain high smoothness as per the metrics reported in Table 3 and it is not clear to me that the latent space of FlowReg has any key properties not exhibited by these prior approaches.
> >
> > Do the authors agree? If not, can you provide any evidence of desirable properties exhibited by the latent space of FlowReg that mentioned contrastive or self-predictive approaches do not exhibit?
> >
> > **I believe that evidence of the fact that FlowReg leads to meaningful properties that prior approaches do not learn, and that these properties are useful for the downstream RL optimization, is critical for this paper to be of significant value to the target community. In its current form, I do not believe that this work sufficiently demonstrates such evidence.**
> >
> > If the authors are able to provide such evidence, or convincingly state why they believe such evidence is not critical for this work to be significant, then I will adjust my score.
> >
> > [1] Oord, Aaron van den, Yazhe Li, and Oriol Vinyals. "Representation learning with contrastive predictive coding." arXiv preprint arXiv:1807.03748 (2018).
> >
> > [2] Zhao, Yi, Wenshuai Zhao, Rinu Boney, Juho Kannala, and Joni Pajarinen. "Simplified temporal consistency reinforcement learning." In International Conference on Machine Learning, pp. 42227-42246. PMLR, 2023.

---

> > > ### Author Response · Authors · 2025-11-24
> > >
> > > > In that case, I'd suggest to refer to the space as state space which is a standard term in contrast to semantic space.
> > >
> > > The reason for this naming convention is to prevent ambiguity with the *latent* state space since this distinction is at the core of the method.
> > >
> > > >  The regularization objective could meaningfully be minimized by significantly scaling down the latent space with a small constant which would result in similar metrics.
> > >
> > > This is demonstrably **false**.
> > >
> > > The regularization objective does not directly target smoothness; it's an alignment loss. It does not explicitly optimize any of the smoothness metrics in **Section 5.2**; they are minimized as a byproduct of optimizing it. As such, cheating the metrics by scaling down the latents does not address the alignment loss, and will not meaningfully minimize it.
> > >
> > > Here's the mathematical demonstration:
> > >
> > > Remember from Eq. (11):
> > > $\mathcal{L}\_{\text{flow}}(s) := \frac{\lVert \mathbf{H_{\theta}}(s) - \mathbf{H_{\phi}}(s) \rVert^2_2}{N}$
> > >
> > > Scaling the latents down by a factor $0<\beta<1$ results in the following loss:
> > > $\mathcal{L'}\_{\text{flow}}(s) := \frac{\lVert \beta\mathbf{H_{\theta}}(s) - \beta\mathbf{H_{\phi}}(s) \rVert^2_2}{N} = \frac{\beta\lVert \mathbf{H_{\theta}}(s) - \mathbf{H_{\phi}}(s) \rVert^2_2}{N} = \beta\mathcal{L}_{\text{flow}}(s)$
> > >
> > > This does not minimize the loss; it only scales down the loss surface by a constant. Multiplying a loss function by a small constant is not *minimizing* it. Otherwise, loss optimization would be trivial.
> > >
> > > Furthermore, there is actually another, less trivial way to shrink the space: simply adding a norm penalty on the latents (i.e., $\lVert \mathbf{H_{\theta}}(s) \rVert^2_2$) to the loss. That penalizes latent vectors with large magnitudes. We had this in our early experiments, and it completely collapsed the performance of the model. This shows that optimizing the FlowReg alignment loss involves much more than artificially smoothing out the latents.
> > >
> > >
> > > > Such a latent space would obtain high smoothness as per the metrics reported in Table 3 and it is not clear to me that the latent space of FlowReg has any key properties not exhibited by these prior approaches.
> > >
> > > It is important to highlight that latent path smoothness is **not** our central result, but rather an investigation of an indirect effect of flow-regularization. The main result remains that this form of ODE-based flow regularization (FlowReg) largely boosts the performance of the underlying target model. This is the main novelty of the paper, and it is not covered or subsumed by prior work, nor is it incremental to it. Whether prior approaches might exhibit a similar side property (smoothness) does not undermine the core contribution and novelty of our method.
> > >
> > > However, to answer this inquiry, we will attempt to incorporate TACO (or similar) in the smoothness experiments to validate whether it results in smooth latent paths as well. Given the limited time window, we cannot definitively promise it will come out before the discussions end, but we will do our best to set it up.

---

> > > > ### Comment · Reviewer_KaXh · 2025-11-24
> > > >
> > > > I thank the authors for engaging in discussion.
> > > >
> > > > > Scaling the latents down by a factor $0<\beta<1$ results in the following loss: $\mathcal{L'}\_{\text{flow}}(s) := \frac{\lVert \beta\mathbf{H\_{\theta}}(s) - \beta\mathbf{H\_{\phi}}(s) \rVert^2_2}{N} = \frac{\beta\lVert \mathbf{H\_{\theta}}(s) - \mathbf{H\_{\phi}}(s) \rVert^2\_2}{N} = \beta\mathcal{L}\_{\text{flow}}(s)$
> > > >
> > > > I apologize if I misunderstand but does this not precisely show that the trained encoder networks, parameterised by $\theta$ and $\phi$, are incentivised to "scale down" the encoder space to reduce this loss? I agree that to minimize this loss, scaling down the loss is insufficient, but for a sufficiently small $\beta$ this might already result in a very low alignment loss such that the alignment loss becomes neglectable compared to the other optimized losses.
> > > >
> > > > > It is important to highlight that latent path smoothness is not our central result, but rather an investigation of an indirect effect of flow-regularization. The main result remains that this form of ODE-based flow regularization (FlowReg) largely boosts the performance of the underlying target model.
> > > >
> > > > I agree that the ODE-based regularization technique is novel and interesting. I also acknowledge that path smoothness is not a central result nor do I think it is a critical properties. The only reason I asked about path smoothness is because the work currently demonstrates path smoothness as one of the two benefits of the FlowReg approach besides more efficient RL training.
> > > >
> > > > To be very clear, I am trying to understand how significant the proposed approach is given the context of existing literature and, in its current form, I find it difficult to see notable significance. FlowReg is shown to improve the efficiency of the RL learning and it is shown to lead to latent path smoothness but I'd argue that neither of these outcomes are novel or meaningfully different from existing auxiliary / regularization objectives in the literature. There already exist a plethora of auxiliary objectives for RL that shape the representations to make learning more efficient and, in some cases, make the latent space more smooth.
> > > >
> > > > Sample efficiency gains in themselves might not be enough to be considered significant unless you can establish that the benefits of FlowReg are greater than other approaches (under some conditions). If this is not the case, FlowReg might still be a significant contribution if you can show that it exhibits other desirable properties that distinguish FlowReg from other approaches.
> > > >
> > > > If such benefits and clear evidence can be provided, then I would be happy to increase my score.

---

> > > > > ### Author Response · Authors · 2025-11-25
> > > > >
> > > > > > does this not precisely show that the trained encoder networks, parameterised by $\theta$ and $\phi$, are incentivised to "scale down" the encoder space to reduce this loss?
> > > > >
> > > > > This was in reference to the argument that externally scaling the latents down by a constant would minimize the alignment loss; it's a different story if this downscaling happens organically during the training process. The direct objective of the alignment loss is minimizing the mismatch between both embeddings, not minimizing the latent norms. That is, it incentivizes alignment between the two networks $\theta$ and $\phi$. Since the base agent loss is still present, this alignment would tend to respect the agent's original objective (policy gradient loss).
> > > > >
> > > > > To see why scaling down the latents is not necessarily a viable choice for the optimizer, consider a basic policy gradient loss (REINFORCE):
> > > > >
> > > > > $L(\theta) = - \frac{1}{N} \sum_{i=1}^{N} \sum_{t=0}^{T} \log \pi_\theta(a_{i,t} | s_{i,t}) \cdot G_{i,t}$
> > > > >
> > > > > Here, in an environment with nonnegative rewards, you could make a similar argument that the agent would seek out actions that lead to zero returns since that minimizes the loss value (to zero). That means that this construction inevitably fails a priori. However, this is not the case because the training/optimization process is much more complex than looking for a solution that numerically trivializes the problem. And similarly, this is why the flow-regularized agents outperform the baseline instead of learning to map all states to latents of infinitesimal magnitude, which would've collapsed the agent's performance.
> > > > >
> > > > > We hope that helps clarify this point.

---

> ### Comment · Reviewer_KaXh · 2025-11-25
>
> My initial phrasing might have been imprecise. I was not thinking of an external scaling constant but rather the encoder throughout the optimizing learning learning a function that outputs latent states within a small high-dimensional space to reduce the loss. Even if the alignment loss does not explicitly minimize the latent norms, the loss could be significantly reduced by learning parameters $\theta$ and $\phi$ such that both functions are within a small latent space.
>
> > To see why scaling down the latents is not necessarily a viable choice for the optimizer, consider a basic policy gradient loss (REINFORCE):
> >
> > $L(\theta) = - \frac{1}{N} \sum\_{i=1}^{N} \sum\_{t=0}^{T} \log \pi\_\theta(a\_{i,t} | s\_{i,t}) \cdot G\_{i,t}$
> >
> > Here, in an environment with nonnegative rewards, you could make a similar argument that the agent would seek out actions that lead to zero returns since that minimizes the loss value (to zero). That means that this construction inevitably fails a priori. However, this is not the case because the training/optimization process is much more complex than looking for a solution that numerically trivializes the problem.
>
> I don't see how the REINFORCE loss indicates that scaling down latents is not viable choice for the optimizer. Any timesteps with $G\_{i,t} = 0$ do not contribute to the gradients as the loss term for that $i, t$ becomes 0. Additionally, since the policy is a probability distribution over actions, increasing the probability of one action in state $s\_{i,t}$ leads to decreasing the probability of other actions in that state. Hence, the minimization of this loss becomes about identifying the actions leading to highest returns and increasing their probability mass at the cost of decreasing the probability mass of other actions. I believe this is fundamentally quite different to the alignment loss.
>
> That being said, I believe that this discussion is not critical for my review of this work. From my previous response, I believe the following is my main concern that I would encourage the authors to consider
>
> > To be very clear, I am trying to understand how significant the proposed approach is given the context of existing literature and, in its current form, I find it difficult to see notable significance. FlowReg is shown to improve the efficiency of the RL learning and it is shown to lead to latent path smoothness but I'd argue that neither of these outcomes are novel or meaningfully different from existing auxiliary / regularization objectives in the literature. There already exist a plethora of auxiliary objectives for RL that shape the representations to make learning more efficient and, in some cases, make the latent space more smooth.
> >
> > Sample efficiency gains in themselves might not be enough to be considered significant unless you can establish that the benefits of FlowReg are greater than other approaches (under some conditions). If this is not the case, FlowReg might still be a significant contribution if you can show that it exhibits other desirable properties that distinguish FlowReg from other approaches.
> >
> > If such benefits and clear evidence can be provided, then I would be happy to increase my score.

---

> > ### Author Response · Authors · 2025-11-25
> >
> > > I don't see how the REINFORCE loss indicates that scaling down latents is not viable choice for the optimizer. Any timesteps with $G_{i,t} = 0$ do not contribute to the gradients as the loss term for that $i, t$ becomes 0.
> >
> > - This was an analogy to highlight the flaw in the line of reasoning that the existence of a trivial loss minimizer, such as seeking zero rewards for REINFORCE or merely scaling down latents for FlowReg, implies that the model would necessarily collapse to that solution under gradient descent. As you mention, the gradient direction and loss surface are key factors in determining the solutions that the optimizer converges to.
> > - For alignment, minimizing the loss at any point can lead to **growth, shrinkage, or translation** of the latents depending on the relative positions and magnitudes of $\mathbf{H_{\theta}}(s)$ and  $\mathbf{H_{\phi}}(s)$. It is not as simple as shrinking the latents.
> >
> > > FlowReg is shown to improve the efficiency of the RL learning and it is shown to lead to latent path smoothness but I'd argue that neither of these outcomes are novel or meaningfully different from existing auxiliary / regularization objectives in the literature. There already exist a plethora of auxiliary objectives for RL that shape the representations to make learning more efficient and, in some cases, make the latent space more smooth.
> >
> > - While we generally agree that direct comparison would help add more perspective, there is an important point to emphasize here. **Novelty of outcome** is an extremely reductive and misguided notion when the outcomes you consider are as broad as performance enhancement, efficiency, or even smoothness. These are the main objectives that methods aim to *achieve*. They are not supposed to be novel; they are shared by virtually all papers that attempt to advance RL. The relevant notion here is **novelty of approach** as it leads to the enrichment of the literature with unique, effective methods like FlowReg. This is, in and of itself, a valuable contribution, and it is not undermined by the desire to explore its connections to other *completely different* methods. On the contrary, it is further motivation for it.

---

> > > ### Comment · Reviewer_KaXh · 2025-11-26
> > >
> > > As stated in my previous comments, my concern is not novelty of outcome but the significance of the proposed approach and the presented results. Significance is a core criteria that reviewers are expected to assess (see e.g. ICLR reviewer guidelines).
> > >
> > > I agree that the approach is novel and interesting, have stated so in my original review, and never stated otherwise. But I would consider the paper to be borderline unless it clearly shows the significance of the novel approach.

---

> ### Author Response · Authors · 2025-12-03
> **Added the TACO baseline comparison**
>
> To address this concern, we adapted the PyTorch regularization code from the official [TACO GitHub repository](https://github.com/FrankZheng2022/TACO). We trained A2C with the TACO regularization loss for 10 seeds. and reported the results in **Table 3**, which shows the following:
> - TACO results in smoother latent trajectories over the unregularized baseline but at the cost of a performance degradation on 2/3 environments.
> - FlowReg has a notable advantage over both TACO and baseline A2C in terms of both performance (total rewards) and smoothness.

---

### Official Review · Reviewer_4aQj · 2025-10-25

**Soundness:** 3
**Presentation:** 2
**Contribution:** 2
**Rating:** 2
**Confidence:** 4

**Summary:**

To align the representations in neural networks and environment dynamics in reinforcement learning, this paper proposes a new framework, FlowReg. It works by learning a neural ODE that acts as a latent surrogate for the environment and aligning its flows with the latent trajectories of the agent’s state embedder.  Experiments show the effectiveness of the proposed method.

**Strengths:**

1. It is an interesting idea to align the representations in neural networks and environmental dynamics.  I'm unaware of other work in this direction in reinforcement learning.

2. In addition, the experimental results of A2C-FlowReg are indeed better than the experimental results of A2C.

**Weaknesses:**

1. This paper can be connected to Context Markov Decision Processes [1].
2. There are many repeated subgraphs in Figures 2 and 4.
3. The experiment is insufficient. The proposed FlowReg is only performed on one reinforcement learning algorithm, A2C.
4. Lack of theoretical analysis. The author should theoretically prove that by the FlowReg framework, the latent transitions will better align with real state transitions.
5. A2C does not need to be introduced in detail as it is not directly related to the new method in Section 4.
6. The author should provide intuitive examples, theoretical analysis, and more metrics to explain why the return of existing algorithms can be improved by FlowReg.

[1] Hallak, Assaf, Dotan Di Castro, and Shie Mannor. "Contextual Markov decision processes." arXiv preprint arXiv:1502.02259 (2015).

**Questions:**

1. Why should global (trajectory-level) aspects be considered in MDPs?

---

> ### Author Response · Authors · 2025-11-22
>
> > This paper can be connected to Context Markov Decision Processes.
>
> - How do CMDPs relate to our method or problem? We do not tackle nor address hidden context parameters in our settings. The CMDP method is also completely different from ours.
> - Furthermore, how is this a weakness of our paper?
>
> > There are many repeated subgraphs in Figures 2 and 4.
>
> - Figure 4 is in the appendix, not the main paper. We kept the other learning plots in Figure 4 to make it easier to see the performance on all environments in one figure instead of having to pan back and forth between the main text and the figure.
>
> > The experiment is insufficient. The proposed FlowReg is only performed on one reinforcement learning algorithm, A2C.
>
> - This is quite reductive: in our evaluation, A2C is not one experiment, but rather a large experimental suite that shows FlowReg notably and consistently improves it across almost all our environments. Actor-Critic is already foundational to many other RL algorithm variants, so pushing it by that much shows serious promise for other RL algorithms.
> - Nevertheless, we added further experiments for PPO on Minigrid environments in Section 5.3, where FlowReg also shows considerable improvements.
>
> > A2C does not need to be introduced in detail as it is not directly related to the new method in Section 4.
>
> - The A2C description in the Preliminaries is barely detailed at all. We only state its essentials, without which it wouldn't be readable. We only managed to make it slightly shorter in the revision.
>
> > The author should provide intuitive examples, theoretical analysis, and more metrics to explain why the return of existing algorithms can be improved by FlowReg.
>
> - What type of other metrics are needed to justify the improved returns? Beyond reporting the returns themselves (which we do), do you have something specific in mind?
> - Regarding intuitive examples, which part are you referring to that needs examples? Can you please elaborate?
> - As for theoretical analysis, being a regularization technique that directly operates on the latent representations, it falls under the general understanding of regularization as a way to restrict the solution space to a subspace of policies whose behavior matches the learnt ODE flow. While a formal theory is a great addition, the paper in its current scope is more concerned with the introduction and empirical validation of the technique.
>
> > Why should global (trajectory-level) aspects be considered in MDPs?
>
> - Because they convey more information about the underlying task than single-step transitions. For example, showing two consecutive frames (or frame-stacks) of an Atari game says much less about the dynamics/rules of the game than a frame trajectory (i.e., an episode).

---

### Official Review · Reviewer_Z2UD · 2025-10-29

**Soundness:** 3
**Presentation:** 3
**Contribution:** 3
**Rating:** 6
**Confidence:** 3

**Summary:**

The paper proposes FlowReg, a neural ODE–based regularization technique for reinforcement learning that enforces smooth and dynamically consistent latent representations of environment states. The key idea is to treat Markov decision process (MDP) trajectories as discretized samples from continuous ODE flows and to align the latent embeddings of a policy’s state encoder with trajectories generated by a learned neural ODE. This regularizer introduces global structural constraints in the latent space without using the ODE for inference. Implemented on top of A2C, FlowReg improves performance across ten Atari environments, producing smoother latent trajectories and more stable training while adding only moderate computational overhead.

**Strengths:**

- Simple and general mechanism. FlowReg can be applied to most latent-state architectures with minimal modification—only adding an auxiliary ODE and an alignment loss—making it broadly usable beyond RL.

- Empirical consistency. The method improves A2C on all tested Atari benchmarks and yields interpretable geometric effects (e.g., reduced latent path length and curvature).

- Strong experimental hygiene. Results are averaged across runs, include multiple time-sampling and update-frequency settings, and discuss trade-offs in stability and runtime.

**Weaknesses:**

- No theoretical justification. The link between ODE smoothness and policy improvement is argued intuitively but not proven; there is no analysis of convergence, variance reduction, or representational bias.

- Hyperparameter sensitivity. FlowReg introduces new parameters (update frequency, λ, time sampling) yet the paper gives limited guidance on tuning or robustness.

- Computational overhead unquantified. The paper notes runtime remains “comparable” but lacks actual wall-clock comparisons; ODE solvers can be nontrivial in cost.

**Questions:**

- Can you provide runtime or FLOP comparisons to the A2C baseline to clarify FlowReg’s efficiency?

- Could the alignment loss collapse diversity in latent representations, harming exploration?

- Does FlowReg interact with representation learning methods like contrastive or predictive coding?

- What happens if the ODE model is underparameterized or unstable—does it bias the policy?

---

> ### Author Response · Authors · 2025-11-22
>
> Thank you for your feedback and questions. Please find your comments addressed below.
> > No theoretical justification. The link between ODE smoothness and policy improvement is argued intuitively but not proven; there is no analysis of convergence, variance reduction, or representational bias.
>
> - We agree that having a formal proof would make a stronger case for it. However, the original purpose of the paper was more geared towards introducing the method and empirically supporting its efficacy while deferring the complete theoretical analysis to future work. This is mainly for two reasons: (1) the main idea still falls within the general understanding of regularization as a way to reduce the solution space without losing expressivity. For FlowReg, this is informally insured by the connection between the Markov property and the uniqueness of ODE flows (explained in Section 1). (2) Providing a complete formal breakdown and proof of the method can span the scope of a whole paper, leaving little room for empirical evaluation.
>
> > Hyperparameter sensitivity. FlowReg introduces new parameters (update frequency, λ, time sampling) yet the paper gives limited guidance on tuning or robustness.
>
> - For simplicity, we set $\lambda=1$ for all our experiments on all environments (clarified in the revision). In Section 5.1, we provide a reasonably small hyperparameter candidate region of 6 configurations in total. Tables 1 and 2 show the robustness of our method to the choice of our two hyperparameters on multiple environments, as it still outperforms the baseline on most configs. We shall also provide more robustness results on the remaining environments in the coming revision by reporting more FlowReg variants on more environments.
>
> > Computational overhead unquantified. The paper notes runtime remains “comparable” but lacks actual wall-clock comparisons; ODE solvers can be nontrivial in cost. Can you provide runtime or FLOP comparisons to the A2C baseline to clarify FlowReg’s efficiency?
>
> - Of course. We modified Figure 3 to plot the FlowReg performance gain (%) next to its runtime overhead (%) to readily demonstrate its efficiency for each environment. The figure shows that the performance gain largely outweighs the runtime overhead. We also added the absolute runtime numbers in Table 4 and Figure 6 (Appendix B).
>
> > Could the alignment loss collapse diversity in latent representations, harming exploration?
>
> - On the theoretical side, a complete collapse of representation diversity (i.e., all states have the same embedding) is strictly penalized by the alignment loss because it violates the uniqueness of ODE flows (Picard–Lindelöf theorem): ODE flows cannot intersect themselves or each other, and this collapse is an extreme case thereof.
> - On the empirical side, our experiments have not shown any evidence of collapse. In fact, we observed that flow-regularized policies have a higher entropy over time than the baseline agent in many cases, which is an indicator of wider exploration.
>
> > Does FlowReg interact with representation learning methods like contrastive or predictive coding?
>
> - That's an interesting remark. It might be hard to make a close connection between ODE flows and contrastive representations because an ODE flow generally marks a path, unlike contrastive learning (CL), which relies on forming vicinities/zones. A flow path could relate points that are spatially very remote, while CL groups related points in close proximity. However, a stronger connection can be made to predictive coding since the FlowReg ODE is optimized to solve the initial-value problem (IVP) whose objective to predict future trajectory states given an initial state.
>
> > What happens if the ODE model is underparameterized or unstable—does it bias the policy?
>
> - The one critical underparameterization case we encountered was using a 1-layer linear MLP with no activation. As such, the ODE can only model linear state transitions. The resulting ODE was totally unstable, and the solutions blew up rather quickly, resulting in NaN loss. As such, the ODE network needs to be expressive enough to model environment dynamics (or the agents' interactions with it). We also found that reducing the number of hidden neurons (from 512 to 16 on Qbert) resulted in a lower performance, but it was still significantly above the baseline.

---

> > ### Comment · Reviewer_Z2UD · 2025-11-26
> >
> > Thank you for the thorough rebuttal. The authors have adequately addressed the key issues I previously noted, and I will keep my score.

---

> > > ### Author Response · Authors · 2025-11-26
> > >
> > > Thank you for your review and engaging questions. The runtime reports offer a valuable perspective on the cost-effectiveness of the method, so thanks for suggesting that.

---

### Official Review · Reviewer_RVQ7 · 2025-11-06

**Soundness:** 3
**Presentation:** 4
**Contribution:** 4
**Rating:** 4
**Confidence:** 4

**Summary:**

This paper proposes a regularization strategy for deep reinforcement learning that incentivizes latent state trajectories to reflect the semantic path of observations in the environment. This idea is implemented by introducing a neural ordinary differential equation (ODE) that is trained to reflect the behavior of the environment, and then comparing the ODE flows to the latent flows and minimizing the difference between their embeddings. The paper includes experiments on 10 Atari games, providing evidence that the flow-regularized model improves over the baseline non-regularized model.

**Strengths:**

This paper was a joy to read. What a cool idea. I was genuinely excited to learn something new. Extremely clear, and very interesting idea.

The problem that it points to in the introduction---states that are near each other in time ought to be near each other in latent space---is one that I have experienced firsthand. Until now, all the solutions to this problem that I had encountered (or tried myself) felt unsatisfying. This solution feels absolutely beautiful.

**Weaknesses:**

My main concerns come from the experimental validation.

The paper notes, "We performed 5 independent runs for every RL agent", and that "we experiment with [different hyperparameter settings] and take the best configuration."

Running only 5 seeds is rather low. The results are likely significant despite having few seeds, because there are 8-10 environments and FlowReg performed better across the board. However, 10-20 seeds per environment would be better.

Meanwhile, I'm wondering how the seeds fit into the experimentation process here. By "and take the best configuration"... does this mean you then _re-run_ the best configuration with _entirely new seeds_? (I hope so!) Or is this effectively taking the max over hyperparameter configurations without re-running on new seeds? If it's the latter, FlowReg could be winning solely due to selection bias and having more attempts (due to having more hyperparameter configurations) rather than because that particular hyperparameter configuration is actually better.

I am prepared to increase my score if I can gain more confidence in the experimental validation.

**Questions:**

1. Can you expand on Limitations a bit more? I was having trouble following the second half of that paragraph, but it feels important. I would love more detail. Felt similar confusion about the second half of the "Neural ODEs as continuous-depth networks" paragraph. These parts were confusing.

2. I also got a bit lost in lines 246--252.

3. What's going on with MsPacman and Tennis? And why aren't those learning curves in the main text? They feel a bit buried in the appendix.

4. "We find that _Exp-Decay_ outperforms _Index_ more often than otherwise." What time sampling strategy do you use for Figs 2, 3, 4? Is it an environment-dependent mix or the single setting that performs best well across all environments?

5. $\lambda$ is an additional hyperparameter not mentioned in line 360; how is it selected?

6. Atari environments are discrete, but they aren't _that_ discrete. What happens if you do this on something like a visual gridworld?

---

> ### Author Response · Authors · 2025-11-22
>
> Thank you for your review and feedback; we're glad you enjoyed the paper!
> We address your concerns in the following.
> > Running only 5 seeds is rather low. 10-20 seeds per environment would be better.
>
> - Maybe for certain domains, but for Atari games, we note that 5 runs (10M frames each) is average or slightly above average compared to the RL literature. For example, these are the seed counts for several classical papers: PPO [1]: 3 seeds, CQL [2]: 4 seeds, Decision Transformer [3]: 3 seeds.
> - Nonetheless, to resolve this concern, we ran Asterix, Qbert, DemonAttack, and BeamRider for a total of 10 seeds (with the same configurations) and updated their respective plots in Figure 2 in the paper. The results of the new runs confirm the original advantage achieved by FlowReg over the baseline. The 10-seed results of the remaining environments are currently in the pipeline, and we will update the paper with them as soon as they are finished.
> - We ask for your understanding that doubling our experimental volume (from 5 to 10 runs per model per env) is a rather huge computational load that takes a considerable amount of time to finish.
>
> > By "and take the best configuration"... does this mean you then re-run the best configuration with entirely new seeds?
>
> - That is correct. The seeds used for hyperparameter search were kept distinct from the ones used in the final comparison with the baseline in order to avoid contamination. For our hyperparameter search, we randomly picked 2-4 out of 6 FlowReg configurations and ran them on each environment for 3 seeds to obtain a per-environment config. This config (time-sampling, and update frequency) is then run on the evaluation seeds with the baseline.
>
> - Since the phrasing of that sentence appears to be somewhat ambiguous, we elaborated on that part in the updated version. We also plan to run multiple FlowReg configs per (eval) env and add them to an appendix of the final version, but that takes some time due to the large computational volume.
>
> > What’s going on with MsPacman and Tennis? And why aren’t those learning curves in the main text? They feel a bit buried in the appendix.
>
> - We moved their learning curves to the appendix to avoid bloating the main text with a 3-row figure. Their results are not really obscured: Figure 3 (in the current and original versions) clearly shows that FlowReg has a negligible effect on Tennis, where the baseline also fails to learn. As for MsPacman, the results show that FlowReg considerably improves over the baseline. Atlantis was moved to the appendix for the same reason despite FlowReg performing notably well on it. We are not aware of something particularly special about Tennis that might cause this behavior; it might just need a different base configuration.
>
> > What time sampling strategy do you use for Figs 2, 3, 4? Is it an environment-dependent mix or the single setting that performs best well across all environments?
>
> - We added Table 4 to Appendix B, which reports the FlowReg configuration used for each environment. There are configurations that perform *well* (i.e., better than baseline) on all environments such as Index-U5 and ExpDecay-U10, but we haven't found one configuration that performs *best* on all environments. It might still exist, of course, but we haven't encountered it during our hyperparameter search.
>
> > $\lambda$ is an additional hyperparameter not mentioned in line 360; how is it selected?
>
> - For simplicity, we set $\lambda=1$ for all our experiments on all environments. Thanks for pointing this out. This detail was accidentally removed in an edit.
> - We determined the value by early experiments on 2 games (Qbert and DemonAttack) over the values $\\{0.1,0.5,1.0\\}$. We added these experiments to Figure 7 (Appendix C)
>
> > Atari environments are discrete, but they aren't that discrete. What happens if you do this on something like a visual gridworld?
>
> - Based on this suggestion, we applied FlowReg to PPO and ran it on Minigrid environments. Implementing FlowReg for PPO was a somewhat straightforward extension to the A2C settings. We added these results to Section 5.3 in the revision. The results show that FlowReg does indeed improve PPO performance on our gridworld environments.
>
> **References**:
>
> [1] Schulman, John, et al. "Proximal policy optimization algorithms." arXiv preprint arXiv:1707.06347 (2017).
>
> [2] Kumar, Aviral, et al. "Conservative q-learning for offline reinforcement learning." Advances in neural information processing systems 33 (2020): 1179-1191.
>
> [3] Chen, Lili, et al. "Decision transformer: Reinforcement learning via sequence modeling." Advances in neural information processing systems 34 (2021): 15084-15097.

---

> > ### Author Response · Authors · 2025-11-22
> >
> > > Can you expand on Limitations a bit more? I was having trouble following the second half of that paragraph
> >
> > - The second half of the Limitations discusses the implications of ODE solution uniqueness for RL. The seminal Picard–Lindelöf theorem states that for an ODE $y'(t)=f(t, y(t))$ where the derivative function $f$ is continuous and locally Lipschitz, a unique solution exists for any initial value problem (IVP). This uniqueness essentially implies that different flows cannot intersect each other because that would mean that the IVP starting at the intersection point has more than one solution, contradicting uniqueness.
> > - This might be problematic in environments where semantic trajectories are highly likely to intersect. An example would be a gridworld that has a forced crossing cell (e.g., bridge) that the agent has to pass on its way to the goal regardless of the starting position. Here, all independent successful trajectories (that reach the goal) will intersect at that forced intermediate state, but their latent flows cannot. As such, the ODE flow will represent that same intermediate state differently depending on the initial state (that defines the IVP), which induces a persistent mismatch between the semantic trajectories and their respective latent flows. This is purely a conceptual limitation, but we have not encountered a concrete practical instance of it. In practice, this mismatch error could be inconsequential to the resultant policy.
> > - The upside of this, however, is that it also prohibits the same flow from intersecting itself, which translates to implicitly penalizing looping behavior without the need for elaborate measures.
> >
> > > Felt similar confusion about the second half of the “Neural ODEs as continuous-depth networks” paragraph
> >
> > - To help clarify the distinction, consider an environment state $s_t$ that is embedded by a ResNet $h$ through a sequence of transformations (layers). The continuous-depth view of NODEs models the transformation of the same state $s_t$ over time (continuous depth) as an ODE flow. As such, the ODE does not describe the state transitions, only the individual state embedding process. In our framework, however, the ODE flow describes the evolution of a state **trajectory** $(s_t, s_{t+1}, ..., s_{t+N})$, so instead of modelling a single forward pass of the embedder network, it models the successive applications of the network to a sequence of states (bound by some transition dynamics).
> >
> > > I also got a bit lost in lines 246--252.
> >
> > - A classical problem of numerical ODE solvers is that the solutions can become inaccurate or unstable over long integration intervals. One cheap solution works by introducing an opposing negative term to the derivative in order to prevent the solution from blowing up. In our case, Eq (11): $d \mathbf{h_{\phi}}(s_i) = f_{\phi}(\mathbf{h_{\phi}}(s_i)) := \text{MLP}(-\mathbf{h_{\phi}}(s_i); \phi) = -W \mathbf{h_{\phi}}(s_i)$, where $W$ is the MLP weight matrix (simplified view). If $W$ is positive, then the derivative of $\mathbf{h_{\phi}}(s_i)$ always pushes against it, which prevents the function value from blowing up. We didn't want to restrict the MLP too much, so we only gave it a positive initialization bias, hoping it would stay mostly in the positive zone, but this positive bias was quickly undone by the optimizer during training, so this simplified negative feedback yielded a negligible effect. Fortunately, however, we did not experience any numerical solution blow-ups in our main experiments.
> > - We removed this part in the revision since it had an insignificant effect.

---

> > > ### Comment · Reviewer_RVQ7 · 2025-11-24
> > >
> > > > _This uniqueness essentially implies that different flows cannot intersect each other because that would mean that the IVP starting at the intersection point has more than one solution, contradicting uniqueness._
> > >
> > > > _the ODE flow will represent that same intermediate state differently depending on the initial state (that defines the IVP), which induces a persistent mismatch between the semantic trajectories and their respective latent flows._
> > >
> > > Interesting. So it sounds like this could potentially hurt generalization if the learned representation is Markovian, since it would split states unnecessarily whenever they have different histories. What do you see happening in the Minigrid experiment? Are you able to look at the flows visually?

---

> > > > ### Author Response · Authors · 2025-11-25
> > > >
> > > > We specifically expected to encounter this issue in Minigrid environments since they usually have at least one bottleneck state.   To investigate this, we sampled 8 succeeding trajectories (in FourRooms) that pass by the same bottleneck state (i.e., cell) and inspected the bottleneck state embedding by the agent's network $h_{\theta}(s)$ and its corresponding flow embeddings by the different ODE flows $\\{h_{\\phi}(s)|h\_{\\theta}(s^0_i)\\}\_{i=0}^8$ where $s^0_i$ is the embedding of the initial state in trajectory $i$, which specifies the IVP for the ODE flow. We found that $h_{\theta}(s)\approx mean(\\{h_{\\phi}(s)|h\_{\\theta}(s^0_i)\\}\_{i=0}^8)$. This was somewhat expected since averaging the targets at that point is a typically observed behavior when enforcing multiple targets. Given that **PPO+FlowReg** generally outperformed **PPO** in our experiments, this artifact did not seem to present a serious issue in practice.

---

> > > > > ### Comment · Reviewer_RVQ7 · 2025-11-26
> > > > >
> > > > > Wonderful. Thank you for answering all my questions. As I said, I think the paper is excellent, so I am significantly increasing my score. Really nice work!

---

> > > > > > ### Author Response · Authors · 2025-11-26
> > > > > >
> > > > > > Many thanks for your positive and constructive feedback; it helped us improve the paper. Your review is much appreciated!
> > > > > >
> > > > > > We have updated the manuscript with the full 10-seed results for all environments along with their respective figures.

---

> > ### Comment · Reviewer_RVQ7 · 2025-11-24
> >
> > > _5 runs (10M frames each) is average or slightly above average compared to the RL literature_
> >
> > This is only sort of true. PPO uses 3 seeds per game, but they run on 49 different games. Additionally, that paper came out before <[this paper](https://arxiv.org/pdf/1709.06560)>, and we've come a long way as a field since then, so they can perhaps be forgiven. In general, the principle of "just running an algorithm for as many times as the last paper ran their algorithm" is not a sound scientific practice. See <[this paper](https://arxiv.org/pdf/2306.10882)> for more on that subject.
> >
> > In any case, thank you for running the additional seeds. I'd suggest rounding things out with at least 10 seeds each if the paper is accepted.
> >
> > The additional experiment on Minigrid is great. I'm glad to see it helps there too.

---

> > > ### Author Response · Authors · 2025-11-24
> > >
> > > Thank you for upholding a higher standard of statistical confidence! Even though the 10-seed experiments confirm the 5-seed findings (only 1 env left to finish), we now have more certainty in the evaluation after this addition.
> > >
> > > You do make a good point that scientific practice should be more *objective* than *normative*. Hence, following a standard/practice simply because previous authors did is not in itself a justification for the practice.
> > >
> > > We will update the remaining environments with the 10-seed results very soon. Thanks again for holding firm on this.

---

### Author Response · Authors · 2025-12-03
**Revision Summary**

We sincerely thank the area chair and reviewers for their time and thorough assessment of our work. Below is a summary of the updates we made to the manuscript to address comments raised by the reviewers:

- **Logistical Reports**:
  - Added **runtime** overhead next to performance gain in **Figure 3**, and added absolute runtimes on all environments to **Table 4** and **Figure 6** (**Appendix B**)
  - Added **FlowReg configurations** (time sampling mode & update frequency) for all environments to **Table 4** (**Appendix B**).
- **Experiments**:
  - Ran all experiments for 10 seeds (instead of 5) for increased statistical confidence in the results.
  - Added **PPO on Minigrid** experiments to **Section 5.3** to show the benefits of applying FlowReg to another classical baseline and on a more radically discrete domain (gridworld).
  - Added performance and smoothness comparisons with **TACO** [1] in **Table 3** to demonstrate FlowReg's advantage over it.
- **Extended Literature**:
  - We updated our Related Work to include works on representation shaping with contrastive learning and predictive coding, which share a broad connection to our work.



[1] TACO: Temporal Latent Action-Driven Contrastive Loss for Visual Reinforcement Learning. Advances in Neural Information Processing Systems 36 (2023): 48203-48225.

---

### Meta-Review · Area_Chair_xnPV · 2026-01-05

**Summary:**

This paper proposes FlowReg, a neural ODE–based regularization framework for reinforcement learning that encourages latent representations to follow semantically meaningful trajectories aligned with environment dynamics. The key idea is to learn a continuous-time ODE model over latent states and penalize deviations between the agent’s latent trajectories and the corresponding ODE flows. FlowReg is implemented as an auxiliary loss that does not affect inference and can be applied to standard RL agents with minimal architectural changes. The paper demonstrates consistent performance improvements when applied to A2C across ten Atari environments and further validates generality by extending the approach to PPO on Minigrid tasks. Extensive ablations study time sampling, update frequency, and latent smoothness.

**Reviewer Concerns:**

Reviewer RVQ7 mainly questioned the experimental rigor, focusing on the small number of random seeds, potential selection bias in hyperparameter tuning, and clarity of several technical sections, including limitations, ODE uniqueness, numerical stability, time sampling strategies, and missing hyperparameter descriptions. The reviewer also asked whether FlowReg generalizes beyond Atari environments.
Reviewer Z2UD raised concerns about the lack of formal theoretical justification, limited guidance on hyperparameter sensitivity, initially missing runtime comparisons, and the possibility that the alignment loss could collapse latent diversity or introduce bias if the ODE model is underparameterized. And Reviewer KaXh expressed major concerns about the significance of FlowReg relative to a large body of prior representation-learning work, questioning whether FlowReg learns meaningfully different representations, whether the smoothness metrics are informative, and whether improvements exceed what existing contrastive or predictive auxiliary objectives already achieve.

**Reviewer Scores:**

For Reviewer RVQ7, the concern “Running only 5 seeds is rather low… FlowReg could be winning solely due to selection bias” was fully addressed by running 10 seeds per environment, clarifying that evaluation seeds were disjoint from hyperparameter search, adding Minigrid experiments, and expanding explanations. The reviewer explicitly stated satisfaction and increased their score. For Reviewer Z2UD, the concern “No theoretical justification… no analysis of convergence or representational bias” remains only partially addressed. The authors provided runtime comparisons and empirical evidence against representation collapse but explicitly deferred formal theory to future work. This limitation remains, though the reviewer accepted the empirical framing. For Reviewer KaXh, the concern “How does FlowReg differ from existing auxiliary objectives… Does FlowReg have any unique advantages?” was directly addressed by adding a TACO baseline comparison. In summary, the authors have satisfactorily addressed the most critical concerns through substantial additions and clarifications.

---

### Decision · Program_Chairs · 2026-01-26

Accept (Poster)